# Graph Convolutional Network: Traffic Speed Prediction Fused with Traffic Flow Data

**DOI:** 10.3390/s21196402

**Published:** 2021-09-25

**Authors:** Duanyang Liu, Xinbo Xu, Wei Xu, Bingqian Zhu

**Affiliations:** College of Computer Science and Technology, Zhejiang University of Technology, Hangzhou 310023, China; ldy@zjut.edu.cn (D.L.); 2111912116@zjut.edu.cn (X.X.); 201806062727@zjut.edu.cn (B.Z.)

**Keywords:** intelligent transportation, traffic speed prediction, spatial–temporal correlation, traffic flow

## Abstract

Traffic speed prediction plays an important role in intelligent transportation systems, and many approaches have been proposed over recent decades. In recent years, methods using graph convolutional networks (GCNs) have been more promising, which can extract the spatiality of traffic networks and achieve a better prediction performance than others. However, these methods only use inaccurate historical data of traffic speed to forecast, which decreases the prediction accuracy to a certain degree. Moreover, they ignore the influence of dynamic traffic on spatial relationships and merely consider the static spatial dependency. In this paper, we present a novel graph convolutional network model called FSTGCN to solve these problems, where the model adopts the full convolutional structure and avoids repeated iterations. Specifically, because traffic flow has a mapping relationship with traffic speed and its values are more exact, we fused historical traffic flow data into the forecasting model in order to reduce the prediction error. Meanwhile, we analyzed the covariance relationship of the traffic flow between road segments and designed the dynamic adjacency matrix, which can capture the dynamic spatial correlation of the traffic network. Lastly, we conducted experiments on two real-world datasets and prove that our model can outperform state-of-the-art traffic speed prediction.

## 1. Introduction

In the field of roadway transportation, traffic speed is an important and fundamental parameter, and it can reflect the dynamic running of road traffic. In practical scenarios, if the traffic speed can be known in advance, it can be used to optimize traffic control effectively and improve traffic efficiency greatly. With accurate values of traffic speed known ahead of time, traffic management authorities can not only reasonably adjust the signal timing of road intersections to guide more vehicles in urban traffic but can also control the number of vehicles at entrances to alleviate the traffic flow of expressways. Nowadays, traffic speed prediction has become an indispensable part of intelligent transportation systems (ITSs) [1], and because of its complexity and dynamics in transportation, it is always of great interest in research and engineering.

Traffic speed prediction is a typical problem associated with spatial–temporal data, especially in expressway or highway scenarios. Expressways are relatively closed and have a high-speed traffic flow, and their traffic data are generally collected from numerous detectors scattered in different locations. The change in traffic speed is not only affected by the vehicles in the target location but is also related to the vehicles converging from the adjacent locations, and the impact of each neighboring location also changes as time goes by. As shown in Figure 1, each node denotes a detection location, and each edge represents the connectivity between locations. Comparing with other neighbors of location ***S***, location ***B*** is the nearest neighbor and has a greater impact on location ***S***, but when there is a traffic jam caused by a collision in location ***B***, the impact of locations ***A*** and ***C*** on ***S*** will be greater than location ***B*** as time goes on. Thus, the spatial–temporal correlation of traffic data is complex and highly dynamic, and it is a great challenge to extract the spatial–temporal correlation from traffic data.

The earliest methods of traffic speed prediction are based on the traditional statistical model, such as historical average (HA), vector auto-regressive (VAR) [2] and auto-regressive integrated moving average (ARIMA) [3,4]. These types of models and their variants require the data to satisfy some statistical assumptions: for example, the mean value and variance are always unchanged. These stationary assumptions are inconsistent with the nonlinear system, and if using such methods in traffic systems, the prediction accuracy is not reliable. In order to flexibly fit the nonlinearity of traffic systems, some dynamical modeling techniques are applied to formulate traffic problems. The Kalman filter model [5,6] can overcome the limitation of the classical statistical models and forecast future traffic information, and the hidden Markov model [7] and Bayesian model [8] are also under research in traffic prediction. Although these methods have made progress, they consume massive computation, and the prediction accuracy is degraded by the assumptions and simplifications among these models. 

With the rapid development of data processing techniques, data-driven methods have drawn more attention. Machine learning methods can extract rules and features from a large amount of traffic data. The k-nearest neighbor (KNN) method was first applied to traffic forecasting in [9] and improved by a sequential search strategy in [10], but it still has to spend substantial costs to find an optimal solution with the continuous growth of historical data. Support vector regression (SVR) is a variant of the support vector machine (SVM), which can map multi-dimensional data into a feature space and perform a linear regression within that space [11]. Some recent studies [12,13,14,15] have proved that SVR is superior to other time series methods. However, these machine learning methods have limitations in high-dimensional stochastic traffic environments, and neural network approaches can outperform machine learning methods due to their capability of fitting nonlinear and processing multi-dimensional data [16]. Van et al. [17] adopted a recurrent neural network (RNN) to model state-space dynamics for freeway travel time prediction; Ma et al. [18] presented RNN’s variant long short-term memory (LSTM) to forecast traffic speed; and Huang et al. [19] and Li et al. [20] proposed a model consisting of a deep belief network (DBN) for traffic flow prediction. However, these neural network models only concern temporal characteristics of traffic data, and the improvement in traffic prediction is under expectations. 

In order to capture both temporal features and spatial dependences from stochastic traffic data, some studies have improved in this area. Lv et al. [21] proposed a deep learning approach to traffic forecasting with a stacked autoencoder (SAE) model and discovered spatial and temporal correlations in traffic data. Zeng et al. [22] incorporated the previous (temporal) inputs and exogenous (spatial) inputs to train a recurrent neural network. Gu et al. [23] extracted spatial features by using entropy-based gray relation analysis and captured the lane-level spatial–temporal characteristics of traffic speed with a two-layer model combining LSTM and a gated recurrent unit (GRU). Liu et al. [24] adopted two attention mechanisms to explore import spatial–temporal information. However, more researchers have applied convolutional neural networks (CNNs) to capture spatial adjacent relations among traffic networks, along with employing RNN or its variants on the time axis. By combining an LSTM network and a CNN, Wu and Tan [25] carried out the first attempt to align spatial and temporal regularities and presented a feature-level fused architecture for short-term traffic forecasting. Du et al. [26], Yu et al. [27] and Yao et al. [28] also integrated CNNs and LSTM to model spatial dependencies and temporal dynamics for traffic prediction. Although the spatial–temporal features of traffic data can be extracted by these models, their inputs must be standard grid data. Meanwhile, RNN-based networks require iterative training, and they are widely known to be difficult to train and computationally heavy. 

Recently, spectral graph theory has realized the graph convolution operation for graph-structured data. A general graph convolution framework was firstly proposed by Bruna et al. [29] and further optimized by Chebyshev polynomial approximation in [30] and first-order approximation in [31]. Compared with CNNs, graph convolutional neural networks (GCNs) can implement a convolution operation on graph-structured data. When applying a GCN in the traffic domain, it can effectively encode road network data and extract spatial information, and thus quite a few GCN models have been studied to solve traffic forecasting problem in recent years. Yu et al. [32] proposed a novel spatial–temporal GCN model by using a complete convolutional structure, and such a model can enable much faster training than regular convolutional and recurrent units. Zhao et al. [33] presented a temporal GCN model consisting of a GCN and a gated recurrent unit. Guo et al. [34] introduced an attention-based GCN model to obtain different periodic temporal properties of the traffic flow. Chen et al. [35] invented bicomponent graph convolution in their GCN model to implement the interactions of both nodes and edges of a traffic network. Song et al. [36] designed a spatial–temporal synchronous modeling mechanism in a GCN model to capture the complex localized spatial–temporal correlations. Huang et al. [37] tackled both long- and short-term traffic prediction tasks in their novel GCN model. Although these models use a GCN to encode the topological structure of the traffic network and extract spatial dependences, they only consider the static spatial topology of roads and ignore the dynamics of spatial dependency, which cannot reflect dynamical spatial–temporal correlations of traffic data in real time. Furthermore, when forecasting traffic speed in the future, most studies only used historical data of traffic speed to train the model. Several studies (DCRNN [38], STSGCN [36], ASTGCN [34], etc.) integrated multiple features but only used them simply as model inputs. According to the way of computing and collecting traffic speed data in practical cases, it always has some errors because of the small amount of sample data. These errors may not be large, but they will affect the accuracy of traffic speed prediction to a certain degree.

To overcome the problems mentioned above, we referred to the full convolutional structure proposed in [32] and designed a novel deep learning model of traffic speed prediction: fusing spatial–temporal graph convolutional network (FSTGCN), which can fuse traffic flow data with traffic speed prediction. In this model, we adopt spatial graph convolution and temporal gated convolution to avoid numerous iterative operations. Meanwhile, in order to make full use of the “coupling relationship” between traffic flow and speed, we predicted traffic flow data and integrated it into the speed prediction process. In this way, we would obtain prior knowledge of the future road state through the traffic forecast value. Furthermore, in order to effectively capture the dynamic spatial correlations of the traffic network, we analyzed the covariance relationship of the traffic flow data between every two locations and designed the dynamic adjacency matrix by fusing the dynamic covariance with its static spatial dependence. Lastly, we conducted some experiments on real-world datasets and verified the effectiveness of the model.

To summarize, the main contributions of this paper are as follows:We present a novel deep leaning model called FSTGCN to forecast the traffic speed. In this model, we adopt the full convolutional structure to capture spatial dependencies and temporal features and integrate the data of traffic flow and speed together to forecast the future traffic speed.We design a dynamic adjacency matrix to extract dynamic spatial correlations effectively. We first encode the dynamics of every two locations by their covariance of the traffic flow data and then combine with the static spatial adjacency obtained from the distance.We conduct numerous experiments on real traffic datasets and verify the effectiveness of the model.

The rest of this paper is organized as follows. Section 2 outlines the preliminary concepts and formulates the problem of traffic speed prediction. Section 3 introduces the model structure of FSTGCN in detail. In Section 4, we describe the experimental results. Finally, we summarize the paper in Section 5.

## 2. Preliminaries

In this section, we introduce the definition and the problem statement involving traffic speed prediction. 

**Definition** **1****(Traffic Networks).**
*In this paper, we define the traffic network as an undirected graph*

G=(V,E)

*, where *

E

* is the edge collection of the road network, *

V

* is the node collection of the road network, and *

$V=(v_{1},v_{2},\cdots ,v_{n})$

*, where *

n

* is the number of nodes, and each node represents a detector on the road segment. The adjacency matrix *

A∈ℝn×n

* is used to represent the connection between nodes.*


**Definition** **2** **(Feature** **Matrix).**
*The feature matrices *

$X^{0}$

* and *

$X^{1}$

* represent the historical data of traffic flow and speed, respectively.*

X0=(X10,X20,⋯,Xn0)∈ℝl×n

* and *

X1=(X11,X21,⋯,Xn1)∈ℝl×n

* denote the traffic flow and speed at all road segments, respectively.*

$X_{i}^{0}={\left(x_{t-l+1,i}^{0},x_{t-l+2,i}^{0},\cdots ,x_{t,i}^{0}\right)}^{T}$

* and *

$X_{i}^{1}={\left(x_{t-l+1,i}^{1},x_{t-l+2,i}^{1},\cdots ,x_{t,i}^{1}\right)}^{T}$

* denote the traffic flow and speed of the *

i

*-th road segment over past *

l

* time slices, respectively.*


**Problem** **1.***Given traffic network *G*and feature matrices *X0*and *X1*, the model *FModel* is trained with the historical traffic data to predict the future traffic speed, as shown in the following*.
(1)Y⌢=FModel(G, X0,X1)*where *Y⌢=(Y⌢t+1,Y⌢t+2,⋯,Y⌢t+T)T∈ℝT×n* denotes the predicted traffic speed of all road segments over the next *T*time slice, and *Y⌢j=(y⌢j,1,y⌢j,2,⋯,y⌢j,n)*denotes the predicted traffic speed of all road segments at timestamp*j.

## 3. Methodology

### 3.1. The Framework of FSTGCN Model

As shown in Figure 2, the model FSTGCN consists of four modules and, in total, has three spatial–temporal convolutional (ST-Conv) blocks. The ST-Conv block was proposed by Yu et al. in [32], and it contains two temporal gated convolution layers and one spatial graph convolution layer in the middle, which can fuse features from both spatial and temporal domains, as depicted in Figure 3. In our model, the ST-Conv block in module A is applied to capture spatial and temporal features of the traffic flow and output the predicted traffic flow. Then, module B concatenates the results of the two temporal gated convolutions for the predicted traffic flow and the historical traffic speed and integrates them as the input of the subsequent spatial graph convolution. In module C, the spatial graph convolution extracts the road spatiality, and the temporal gated convolution further captures the temporal features. Although slightly different from Figure 3, models B and C working together can implement the ST-Conv block functionally. Lastly, module D stacks an ST-Conv block to deeply extract the spatiality–temporality and outputs the forecasting results of the traffic speed through an output layer. Furthermore, we consider the nodes’ neighboring relationship by using the change in the traffic flow and design the dynamic adjacency matrix, which is applied in all spatial graph convolutions of this model. 

### 3.2. Fusing with Traffic Flow Data

#### 3.2.1. Graph Convolution with Dynamic Adjacency Matrix

The core of graph convolution is Laplacian eigendecomposition, and the Laplacian is constructed by the degree matrix and adjacency matrix. In the road traffic field, many GCNs use the spatial connectivity to represent the nodes’ adjacency, and some GCNs adopt the exponential distance to denote the spatial correlation. All these methods only consider the static spatiality of road segments, but as shown in Figure 1, their spatial dependency is dynamic in traffic applications. Traffic flow is the basic traffic parameter, and the traffic flow of neighboring road segments has the same trend. Therefore, in this section, we adopt the change in the traffic flow to design the dynamic adjacency matrix of the graph convolution, in order to capture the dynamic spatial relationship of road segments.

The road network is regarded as an undirected graph, and each node represents a detector on the road segment. The traditional convolution cannot directly act on the non-Euclidean road network, and spectral graph theory realizes the graph convolution operation by converting the convolution in the space domain to the product in the frequency domain. Bruna et al. [29] firstly proposed the spectral neural network, and Kipf and Welling [31] defined a layer-wise linear formulation by stacking multiple localized graph convolutional layers with the first-order approximation of the graph Laplacian. The formulation can be written as
(2)Θ∗G x=Θ(L)x     ≈θ(In+D−12AD−12)x      ≈θ(D˜−12A˜D˜−12)x
where Θ∗G denotes the graph convolution, Θ is the spectral kernel, θ is the shared parameter of the kernel, L is the normalized Laplacian where L=In−D−12AD−12, In is an identity matrix, D is the diagonal degree matrix where Dii=∑jAij, A˜=A+In, and D˜ii=∑jA˜ij. The graph convolution for 2D variables is denoted as “Θ∗G X”, with the kernel Θ∈ℝK×Cin×Cout, K is the convolution size, and Cin and Cout are the number of convolution input and output channels, respectively. 

The adjacency matrix in traditional graph convolution mainly has two variants, as shown in the following, and it uses the static road topology information to represent the spatiality of road segments.

**Neighbor adjacency matrix:** The adjacency matrix is defined according to the connectivity of neighboring road segments. This method determines whether the road segment is connected with the neighbors or not, which is as follows
(3)$$A_{ij}^{N}=\{\begin{array}{c} 1,\;v_{i}\;\mathrm{and}\;v_{j}\;\mathrm{is}\;\mathrm{adjacent}\\ 0,\;\mathrm{otherwise}\;\;\;\; \end{array}$$

**Distance adjacency matrix:** This method uses the exponential distance to analyze the spatial correlation between road segments [38]. These types of connectivity are induced by roads such as motorways and expressways. Its connectivity is defined as
(4)$$A_{ij}^{S}=\{\begin{array}{c} \exp\left(-\frac{{\left\|x_{i}-x_{j}\right\|}^{2}}{\sigma^{2}}\right),\;A_{ij}^{S}\geq \varepsilon \;\\ 0,\;\;\;\;\;\;\;\;\;\;\mathrm{otherwise} \end{array}$$
where ε is used to control the sparsity of the matrix, ${\left\|x_{i}-x_{j}\right\|}^{2}$ is used to calculate the distance between road segments i and j, and $\sigma^{2}$ is the spatial attenuation length. In the experiments, $\sigma^{2}$ and ε were set as 100 and 0.5.

As shown in Figure 1, the adjacency matrix obtained by static coding is defective, and although it describes the static spatial correlation between road segments, it cannot reflect the dynamic change in road segments in real time. The change in the traffic flow on road segments will affect the correlation between them, and thus we introduced the traffic flow to the spatiality of road segments and designed the dynamic adjacency matrix.

**Dynamic adjacency matrix:** When the traffic flow of two road segments changes in a similar way, this can be regarded as adjacency. Thus, we can use the covariance of the traffic flow to calculate the dynamic spatial correlation between road segments, and the covariance adjacency matrix is as follows
(5)$$A_{ij}^{C}=\frac{1}{l-1}\sum_{t=1}^{l}\left(x_{i,t}^{0}-{\bar{x}}_{i}^{0})(x_{j,t}^{0}-{\bar{x}}_{j}^{0}\right)$$
where ${\bar{x}}_{i}^{0}$ and ${\bar{x}}_{i}^{0}$ are the average traffic flow of road segments i and j over the past l time slices, respectively, that is, ${\bar{x}}_{i}^{0}=\frac{1}{l}\sum_{t=1}^{l}{\bar{x}}_{i,t}^{0}$ and ${\bar{x}}_{j}^{0}=\frac{1}{l}\sum_{t=1}^{l}{\bar{x}}_{j,t}^{0}$.

Then, considering the static spatiality, the dynamic adjacency matrix is designed as the Hadamard product of the distance adjacency matrix and the covariance adjacency matrix, which is as follows
(6)$$A_{ij}^{D}=A_{ij}^{S}*A_{ij}^{C}$$

As shown in (6), the exponential distance can decide the static spatial adjacency between two nodes, and the covariance of the traffic flow can determine their dynamic spatial adjacency. Thus, if both their covariance and distance correlation are large, the two nodes will influence each other to a large extent.

#### 3.2.2. Fusing in the Modules

When using neural networks, many researchers only apply historical data of traffic speed and capture valuable information to forecast the traffic speed in the future by training and testing the model. Several studies ([34,36,38], etc.) integrated multiple features but only used them simply as model inputs. In practical applications, there are some errors in not only the collection but also the computation of traffic speed. Due to the small amount of sampled vehicles, computing error is inevitable. Although these errors are not large, they will influence the prediction accuracy of traffic speed to a certain extent. Comparing with traffic speed, the data of traffic flow are more accurate, and multiple techniques can obtain the exact number of vehicles in a period, such as loop detectors and video detectors. Thus, considering the relationship between traffic flow and speed, we used module A to predict traffic flow information, and took it as one of the inputs of module B. In this way, we would obtain prior knowledge of the future road state through the traffic forecast value. 

In traffic theory, both traffic flow and speed are the basic characteristic parameters, and their relationship is
(7)$$x_{i}^{0}=k_{i}x_{i}^{1}$$
where $k_{i}$ is the vehicle density at timestamp i. Additionally, according to the model of traffic flow and speed proposed by Greenshield in [39], the relationship can be further derived as
(8)$$x_{i}^{0}=k_{jam}x_{i}^{1}(1-x_{i}^{1}/x_{f}^{1})$$
where $x_{f}^{1}$ is the traffic speed when the number of vehicles is small and every vehicle can travel at a free speed, and $k_{jam}$ is the vehicle density when the traffic network is seriously blocked. Additionally, the relationship between traffic flow and speed is illustrated in Figure 4. 

As shown in Figure 4, if the traffic speed equals a certain value, i.e., the critical value $x_{cr}^{1}$, the traffic flow will reach the maximum value, i.e., $x_{\max}^{0}$. When the traffic speed is less than the critical value, the traffic flow will increase with the increase in the traffic speed, and the traffic is in congestion. When the traffic speed is larger than the critical value, the traffic flow will decrease with the increase in the traffic speed, and the traffic is free. That is, the traffic flow will be mapped to a certain traffic speed when the traffic is in a certain state. Therefore, we introduced the predicted data of traffic flow into the model. As shown in Figure 2, module A adopts an ST-Conv block to forecast the traffic flow in the future, and the output is
(9)h0=Γ1∗Τ ReLU(gθ0∗G (Γ0∗T X0))
where h0 is the output future traffic flow, X0 is the input historical traffic flow, Γ0∗T and Γ1∗T are the two temporal convolutions, Γ0 and Γ1 are their temporal kernels, gθ0∗G is the graph convolution, gθ0 is the spectral kernel, and ReLU(⋅) denotes the rectified linear unit function. As it was introduced in [32], the temporal convolution layer in the ST-Conv block uses gated convolution to capture the temporality of traffic data, and this method utilizes CNN’s advantages of fast training and simple structures and effectively reduces the complexity of model training. The gated convolution contains 1D causal convolution with a width Kt kernel followed by a gated linear unit (GLU) as the activation function, and Kt is the convolution size. Without filling, each gated convolution will shorten the length of the input by Kt−1. Therefore, the input is X0∈ℝl×n×Cin (in this case, Cin=1) and the output is h0∈ℝ(l−2(Kt−1))×n×Cout.

Then, the temporal convolution in module B concatenates the output of module A and the historical data of traffic speed, which is shown in the following.
(10)h1=Γ2∗T X1⊕ Γ3∗T h0
where h1∈ℝ2l−4(Kt−1)×n×Cout is the output result of module B, ⊕ is the concatenation operator, and Γ2 and Γ3 are the two temporal kernels. The output h1 is the concatenated result of the temporal convolution of the input data X1 and h0.

Subsequently, the output of module B is processed by the graph convolution and temporal convolution in module C as follows.
(11)h2=Γ4∗T ReLU(gθ1∗G h1) 
where h2∈ℝ2l−5(Kt−1)×n×Cout is the output of module C, and gθ1 and Γ4 denote the spectral kernel and the temporal kernel, respectively. As shown in Figure 2, modules B and C together represent an ST-Conv block, and this block also includes two temporal convolutions and one graph convolution. 

Finally, module D continues to handle the result in (11) by using an extra ST-Conv block and output the forecasting values of the traffic speed with the output layer. The formulas are
(12)h3=Γ6∗T ReLU(gθ2∗G (Γ5∗T h2))
(13)Z=Γ7∗T h3 
(14)Yt+j=Zw+b j∈[1,2,⋯,T]
where (12) denotes the processing of the ST-Conv block in module D and its output result is h3∈ℝ2l−7(Kt−1)×n×Cout, gθ2 is the spectral kernel, and Γ5 and Γ6 are the temporal kernels. The output layer comprises an extra temporal convolution and a fully connected layer, and (13) and (14) denote the processing of the temporal convolution and the fully connected layer, respectively. Z∈ℝn×Cout is the result of the extra temporal convolution, and Γ7∈ℝ2l−7(Kt−1)×Cin×Cout is the temporal kernel. Yt+j is the forecasting value of the traffic speed at timestamp j, w∈ℝCout denotes the weight vector, and b is the bias. When forecasting the traffic speed, the first predicted value is Yt+1, and then this value is spliced with the input data; the same process is repeated to forecast the second predicted value until all values are predicted over the next T moments. 

### 3.3. Model Training

When training the model, the root mean square error is employed as a loss function to measure the difference between the predicted result ${\hat{Y}}_{i}$ and the ground truth value $Y_{i}$, which is
(15)$$loss=\sqrt{\frac{1}{n}\sum_{i=1}^{n}({\hat{Y}}_{i}-Y_{i}{)}^{2}}$$

Furthermore, the Adam optimizer is applied, and residual connections are implemented among stacked temporal convolutional layers. 

## 4. Experiments 

### 4.1. Dataset Description

We evaluated our model on two real datasets, PeMSD4 and PeMSD8. Our traffic datasets were collected by the California Transportation Agencies (CalTrans) Performance Measurement System (PeMS). These traffic data contain traffic flow, average lane occupancy and average speed. The details of the datasets are shown in Table 1.

### 4.2. Experimental Settings

The two datasets set the time interval of traffic data to 5 min. Therefore, each node contains 288 data per day. We used the Z-score method to normalize the data to allow the mean of the original data to be 0 and the variance to be 1. In addition, the first 60% of the data were used as the training set, then 20% were used as the verification set and the remaining were used as testing.

The model FSTGCN was implemented based on the Pytorch framework. The size of all graph convolution kernels and temporal convolution kernels in the ST-Conv blocks was set to 3, i.e., K=3 and $K_{t}=3$. The channels of different layers in the ST-Conv blocks were 64, 16 and 64, respectively. 

We used three metrics to evaluate the prediction performance of the FSTGCN model:
Mean Absolute Error (MAE):
(16)$$\mathrm{MAE}=\frac{1}{n}\sum_{i=1}^{n}\left|{\hat{Y}}_{i}-Y_{i}\right|$$Root Mean Square Error (RMSE):
(17)$$\mathrm{RMSE}=\sqrt{\frac{1}{n}\sum_{i=1}^{n}({\hat{Y}}_{i}-Y_{i}{)}^{2}}$$Mean Absolute Percentage Error (MAPE):
(18)$$\mathrm{MAPE}=\frac{100\%}{n}\sum_{i=1}^{n}\left|\frac{{\hat{Y}}_{i}-Y_{i}}{Y_{i}}\right|$$
where n is the number of prediction results ${\hat{Y}}_{i}$, and $Y_{i}$ is the ground truth value.

### 4.3. Baselines

The proposed FSTGCN method was compared to several traditional statistics-based methods and recently proposed GNN-based models, which include the following:Historical average (HA), which uses the average traffic information in the historical periods as the prediction.Vector auto-regressive (VAR), which is the multi-variable extension of the auto-regressive model which can model the correlation between nodes.Support vector regression (SVR), which is one type of machine learning method, and a linear kernel function was used in the experiments.Long short-term memory (LSTM), which is an extension of RNN and has an input gate, a forget gate and an output gate to deal with the long-term dependency and gradient vanishing and explosion problems.STGCN, which applies the full convolutional structure [32] to analyze the spatial–temporal dynamics. The model uses multiple ST-Conv blocks to predict traffic data.ASTGCN, an improved scheme [34] of STGCN, which uses an attention mechanism and three temporal modeling methods to capture dynamic spatial–temporal correlations.

All experiments were trained and tested on the same Linux server (CPU: Intel (R) Xeon (R) Silver 4110 CPU @ 2.10GHz, GPU: NVIDIA Corporation GM204GL (Quadro M4000)) with the existing published implementation.

To guarantee the fairness of the experiments, all approaches were trained by the Adam optimizer, which updates the parameters with a gradient descent algorithm. Additionally, the batch size and the learning rate were set as 50 and 0.001, respectively. All the experiments used historical 60 min data as input to predict the content of the next 60 min.

### 4.4. Experiment Results and Analysis

Table 2 illustrates the prediction results of FSTGCN and baselines at four timestamps (i.e., 15, 30, 45 and 60 min) on datasets PeMSD4 and PeMSD8, and the mean absolute error (MAE), the mean absolute percentage error (MAPE) and the root mean square error (RMSE) are employed to evaluate the prediction accuracy. As shown in Table 2, our proposed model achieves the best performance in both the MAE and MAPE evaluation metrics. Only in dataset PeMSD4 is the RMSE of our model, FSTGCN, larger than that of LSTM. 

Although the performance of the machine learning method (SVR) is beyond expectation, due to the accumulation of errors and the lack of spatial–temporal analysis, the errors increase greatly when the prediction range is expanded. The LSTM model only considers the temporal correlation of traffic data, and its prediction results are not ideal, sometimes even not better than the traditional statistical prediction methods. Comparing with other methods, GCN models obviously outperform, and this proves the effectiveness of spatial graph convolution analysis in the prediction. Meanwhile, our model, FSTGCN, is also better than other GCN models, and this verifies the advantage of our improvement by fusing with the data of the traffic flow.

Figure 5 further investigates the performance of all the above methods in 60 min on the two datasets. As shown in Figure 5, it is obvious that the prediction error of each model is always on the rise with the increase in the time span. The model FSTGCN achieves the best results in short-term prediction, which indicates that the structure of the ST-Conv block can fully capture the spatial–temporal pattern from the traffic data. With the time increasing, the prediction error of our model grows more slowly than the other methods, and this is because the model fusing with the data of the traffic flow can fix the inaccuracy of the predicted traffic speed to a certain degree. Both the dynamic adjacency matrix and the predicted traffic flow as inputs can decrease the prediction error of our model. Therefore, our model has more advantages in the prediction of both the short term and medium–long term.

In particular, in order to verify the advantage of our improvement by fusing with the data of the traffic flow, we designed a new experiment to further illustrate this. In this part, we added Gaussian noise generated by Python’s random distribution function to test sets PeMSD4 and PeMSD8 to enhance the inaccuracy of the speed. As shown in Table 3, when the speed data are more inaccurate, our model, FSTGCN, can effectively utilize the traffic flow data to improve the prediction performance. 

Figure 6 further shows the performance of our model, FSTGCN, and STGCN in 60 min on the two new test sets. The model FSTGCN reduces the MAE by at least 18% on test set PeMSD8. It is obvious that our model, FSTGCN, achieves a good improvement.

Figure 7 presents an example from the PeMSD4 dataset, which illustrates the distribution of different adjacency matrices as described in Section 3.2. The rows and columns in the subfigures represent different nodes, and the color scale bar on the right represents the spatial correlation between nodes by the depth of color. As shown in Figure 7b, the color depth is different in the figure, and it shows that the covariance between nodes is different. Additionally, comparing with the distance adjacency matrix, the dynamic adjacency matrix significantly changes because of its covariance.

In order to evaluate the importance of the predicted traffic flow, we deleted module A from the proposed model and designed FSTGCN (1), which does not fuse the traffic flow data into the input of module B. The results of the experiments on the two datasets are shown in Figure 8. The input of the predicted traffic flow slows down the growth of the prediction error and significantly enhances the prediction performance. With the expansion of the prediction range, this part of the input becomes an important factor to reduce the error.

Except for the predicted traffic flow in module A, the dynamic adjacency matrix fusing with the traffic flow data also benefits the prediction accuracy of the traffic speed. As shown in Figure 9, three different adjacency matrices are applied in the model FSTGCN on the two datasets. Similar results are obtained in that the dynamic adjacency matrix is better than the other two matrices, and its prediction error grows more slowly with the increase in the prediction time. When the traffic flow of the road segments changes in a large range, the dynamic adjacency matrix will be more obvious and effective.

## 5. Conclusions and Future Work

In this paper, we proposed a novel deep learning model of traffic speed prediction: fusing spatial–temporal graph convolutional network (FSTGCN). This model uses the full convolutional structure as its fundamental framework, which applies graph convolution and temporal gated convolution to capture the temporal and spatial characteristics of traffic data, respectively. Then, considering the accuracy of the traffic flow data, we fused it into the model by using an ST-Conv block and concatenated the predicted traffic flow with the historical traffic speed as the input of the forecasting module, thereby correcting the inaccuracy in traffic speed prediction to a certain degree. Meanwhile, we fused the traffic flow data into the dynamic adjacency matrix. We adopted the covariance of the traffic flow data to represent the nodes’ dynamic spatial features, combined the covariance with the static distance and designed the dynamic adjacency matrix. Comparing with other matrices, the dynamic adjacency matrix can reflect the nodes’ adjacency relationship more realistically. Lastly, experiments on two real-world datasets demonstrated that our model, FSTGCN, can outperform other models for traffic speed prediction and verified the effectiveness of our improvements.

In the future, we will further optimize the network structure of the model through an attention mechanism, POI data mining and other strategies to further improve the prediction ability of the model.

## Figures and Tables

**Figure 1 sensors-21-06402-f001:**
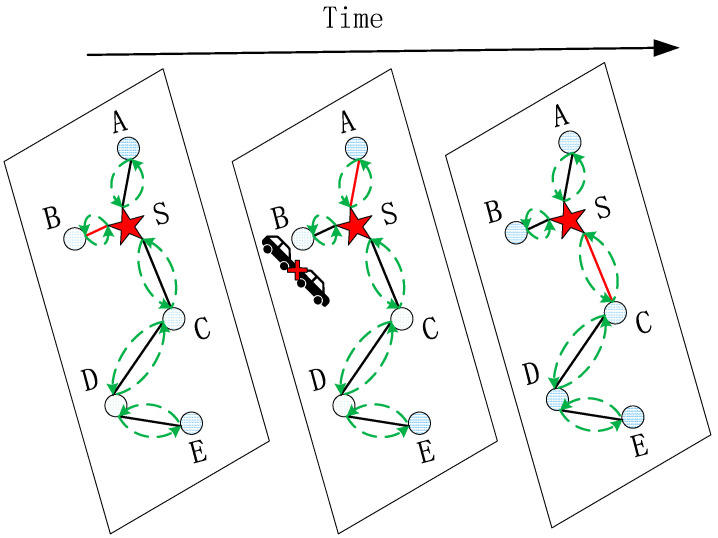
The dynamic spatial–temporal correlation of traffic data.

**Figure 2 sensors-21-06402-f002:**
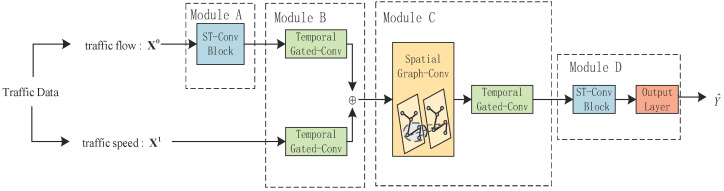
The framework of the FSTGCN model.

**Figure 3 sensors-21-06402-f003:**
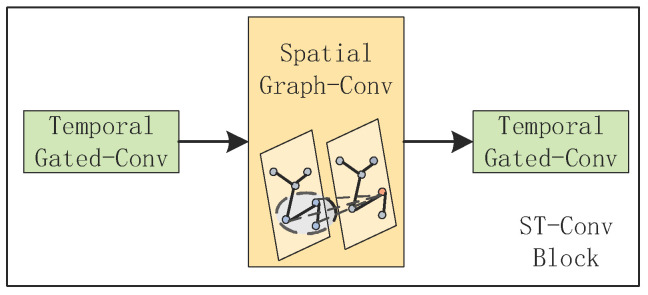
The framework of the ST-Conv block.

**Figure 4 sensors-21-06402-f004:**
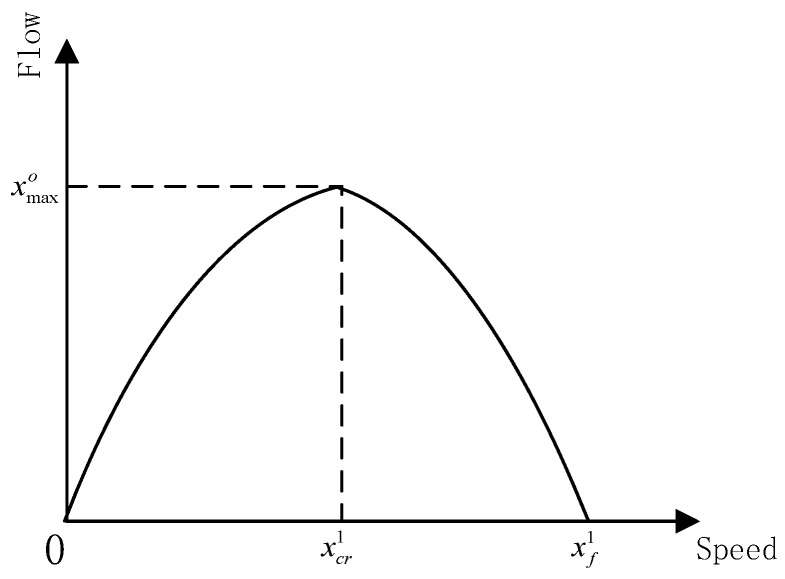
The relationship of traffic flow and speed.

**Figure 5 sensors-21-06402-f005:**
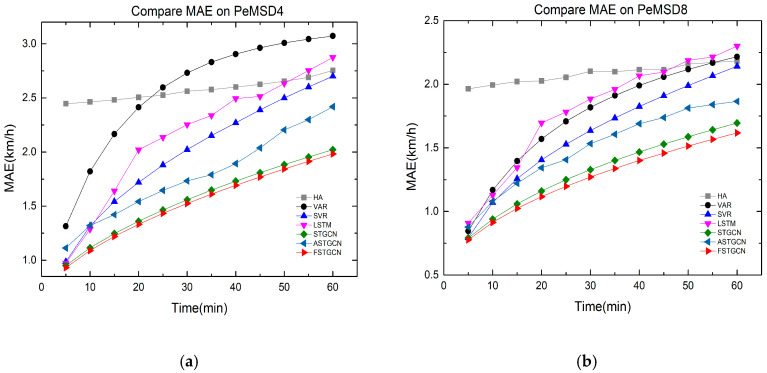
Performance comparison of various methods. (**a**) PeMSD4. (**b**) PeMSD8.

**Figure 6 sensors-21-06402-f006:**
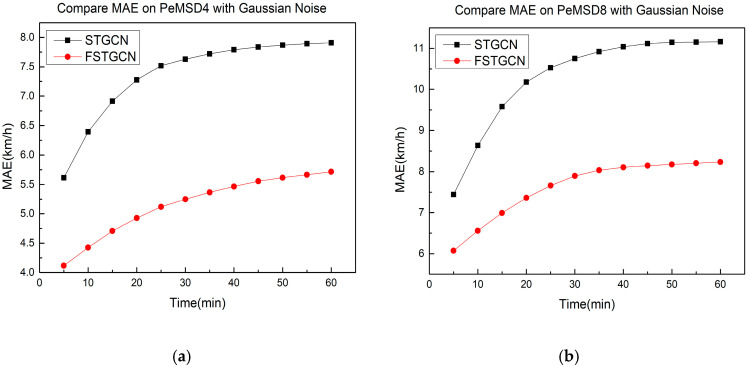
Further comparison on test sets with Gaussian noise. (**a**) PeMSD4. (**b**) PeMSD8.

**Figure 7 sensors-21-06402-f007:**
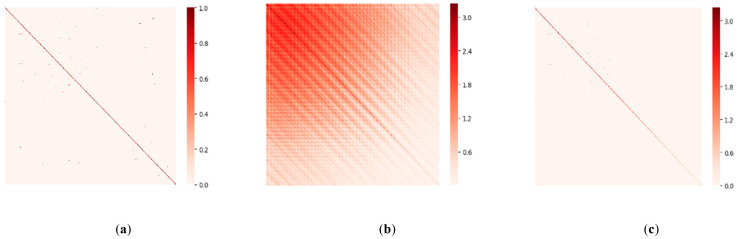
Different adjacency matrices in PeMSD4. (**a**) Distance adjacency matrix. (**b**) Covariance adjacency matrix. (**c**) Dynamic adjacency matrix.

**Figure 8 sensors-21-06402-f008:**
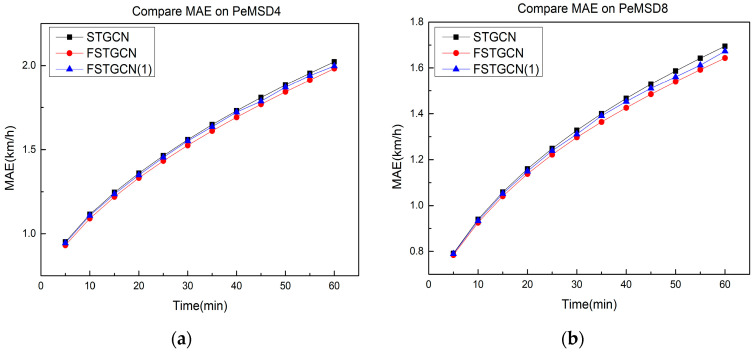
Evaluation of the predicted traffic flow. (**a**) PeMSD4. (**b**) PeMSD8.

**Figure 9 sensors-21-06402-f009:**
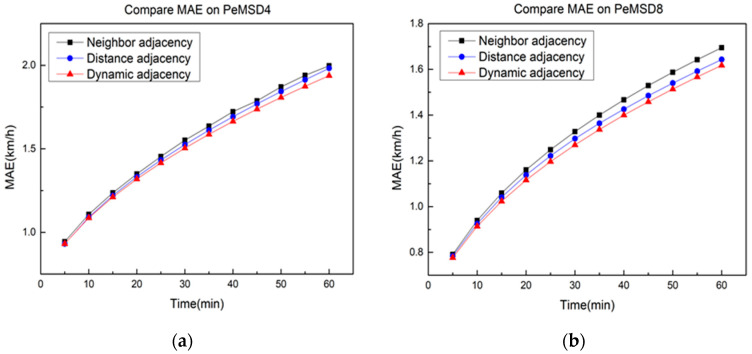
FSTGCN with different adjacency matrices. (**a**) PeMSD4. (**b**) PeMSD8.

**Table 1 sensors-21-06402-t001:** Details of the datasets.

Dataset	PeMSD4	PeMSD8
Timespan	1/2018~2/2018	7/2016~8/2016
Time interval	5 min	5 min
Nodes	307	170
Node features	Flow, occupy, speed	Flow, occupy, speed

**Table 2 sensors-21-06402-t002:** Performance comparison of different methods on PeMSD4 and PeMSD8.

**Model**	**PeMSD4(15/30/45/60 min)**		
	**MAE (km/h)**	**MAPE (%)**	**RMSE (km/h)**
HA	2.446	5.355	5.123
VAR	2.166/2.731/2.963/3.072	4.163/5.401/5.943/6.221	3.651/4.725/5.185/5.404
SVR	1.542/2.022/2.389/2.701	2.935/4.057/4.957/5.731	3.214/4.503/5.426/6.160
LSTM	1.640/2.254/2.518/2.873	2.470/3.559/3.675/4.974	**2.224/3.306/3.391**/4.389
STGCN	1.246/1.559/1.809/2.102	2.514/3.370/4.085/4.687	2.552/3.366/3.977/4.460
ASTGCN	1.421/1.733/1.938/2.419	3.016/3.804/4.348/4.738	2.776/3.533/3.912/4.378
FSTGCN	**1.220/1.525/1.769/1.982**	**2.332/3.060/3.621/4.204**	2.462/3.246/3.835/**4.304**
**Model**	**PeMSD8(15/30/45/60 min)**		
	**MAE (km/h)**	**MAPE (%)**	**RMSE (km/h)**
HA	1.963	4.535	4.656
VAR	1.396/1.818/2.058/2.216	3.236/4.510/5.793/6.012	2.700/3.576/4.053/4.342
SVR	1.258/1.636/1.910/2.141	2.315/3.118/3.728/4.251	2.693/3.718/4.415/4.969
LSTM	1.344/1.844/2.096/2.300	2.334/3.442/4.252/4.678	2.317/3.441/3.517/4.191
STGCN	1.059/1.328/1.529/1.752	2.168/2.845/3.336/3.855	2.280/3.016/3.522/4.082
ASTGCN	1.221/1.533/1.738/1.865	2.721/3.504/4.140/4.578	2.476/3.233/3.612/3.92
FSTGCN	**1.041/1.297/1.485/1.643**	**2.088/2.704/3.134/3.483**	**2.273/3.011/3.512/3.914**

**Table 3 sensors-21-06402-t003:** Further comparison by adding Gaussian noise on test sets PeMSD4 and PeMSD8.

**Model**	**PeMSD4(15/30/45/60 min)**		
	**MAE** **(km/h)**	**MAPE (%)**	**RMSE** **(km/h)**
STGCN	6.915/7.629/7.837/7.909	11.512/13.550/14.111/14.396	9.651/10.404/10.832/11.019
FSTGCN	4.307/4.147/4.112/4.121	7.910/8.078/8.299/8.506	6.319/6.475/6.653/6.800
**Model**	**PeMSD8(15/30/45/60min)**		
	**MAE** **(km/h)**	**MAPE (%)**	**RMSE** **(km/h)**
STGCN	9.581/10.752/11.116/11.162	15.539/17.510/18.163/18.221	14.218/15.772/16.116/16.336
FSTGCN	6.991/7.894/8.145/8.235	11.589/13.078/13.505/13.693	10.147/11.463/11.891/12.131

## Data Availability

The data will be released on https://github.com/Etherealr/FSTGCN.

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
