# Peer review of "Graph Convolutional Network: Traffic Speed Prediction Fused with Traffic Flow Data"

_sensors, 2021, doi:10.3390/s21196402_

Round 1

Reviewer 1 Report

The paper "Graph Convolutional Network: Traffic Speed Prediction Fusing 2with Traffic Flow Data" describes the extension of the ST-GCN model for traffic speed prediction. The authors created a novel approach for generating a dynamic adjacency matrix based on the covariation of traffic characteristics between different roads, as well as a modified architecture of the neural network, fusing the traffic flow rate and speed. The authors have tested their approach on two months of historical PeMS data and have shown minor improvements in the prediction accuracy vs. other compared methods.

There are a few issues with the paper:

  • Moderated editing of language and style is required.
  • Minor typesetting problems like improperly formatted references (line 231) and headers (line 259).
  • Authors claim that "most researches only apply the historical data of traffic speed and capture valuable information to forecast the traffic speed in the future..." (line 261, similar claim in line 122). While it's hard to claim what "most" researchers do, in this reviewer's experience, using traffic flow data as model input, in addition to speed, is quite common.
  • The authors describe a road network as an "undirected graph and each node represents a particular road segment" (line 208). This requires further elaboration of how different traffic directions on a single road were represented.
  • The train/test split is not fully described. The percentage of the split is given, but how exactly the split was performed is not described (was each time frame randomly assigned to train/val/test, or were the first 60% of data designated as the train, then 20% as val, and the last 20% as test?).
  • No description is given of how the "baseline" prediction methods were implemented. Did authors reimplement them (in which case more implementation details are required) or used an existing published implementation (in which case a link to the code would be helpful).
  • No units are specified for the presented accuracy data.

The last point makes it hard to properly evaluate the practical contribution of the paper. If we assume that, like in most other papers on the topic, the error is reported in km/h (or mph), then the proposed model improves MAE by less than 0.05 km/h compared to ST-GCN on the 45-minute prediction horizon. Such an improvement is, in general, negligible for most applications. Furthermore, as the authors correctly mentioned in the paper, measuring traffic speed accurately is challenging. For example, in a study performed by the PeMS research team (https://people.eecs.berkeley.edu/~varaiya/papers_ps.dir/gfactoritsc.pdf), we can see that the speed estimation algorithm for single-loop detectors might have an error over 5 mph, which makes 0.1 mph (or km/h) improvements in accuracy questionable. Dual-loop detectors offer higher-quality data, but they, in general, comprise the minority of all the detectors in the PeMS system.

What can be improved:

  • A more detailed error analysis, showing not only MAE and RMSE. While the 0.1 km/h improvement in mean error is not much for most practical purposes (and it's even questionable if such small error is meaningful given the ground truth data accuracy), the model might, for example, have a lower maximum error, or reliably achieve <1km/h error for all input data. With MAE already below 2 km/h, having tighter boundaries for the worst-case prediction scenario is likely to be more beneficial than further improvements in mean accuracy.
  • An overview of traffic sensor precision is needed. Given such a small difference between the compared models, it is natural to discuss the accuracy of the ground truth data.
  • Testing on only two months of data is not much, and, given the seemingly arbitrary choice of timeframes and PeMS districts, makes it look like a data cherry-picking (although there are likely good reasons for the choice made). Larger intervals (hopefully identical over all the districts) would make the results more convincing and significant.
  • Publishing the code is always welcome. It helps the practitioners to apply the model and helps other researchers to study and improve it.

In conclusion, while the paper claims that improvements were made compared to previous State-of-the-Art methods, the presented results barely support this claim. The reduction in the prediction error is very minor (likely below the accuracy of the PeMS data), shown on a small data set, and the baseline methods are very poorly described. The proposed improvements, in particular the dynamic adjacency matrix, are interesting, but by themselves are not of great significance, and the observed results do not convincingly demonstrate their benefits.

A more thorough benchmarking might help to evaluate the methods and show the effect of the proposed improvements to the baseline ST-GCN. But in the current form, the authors' findings do not show any noticeable improvements of their methods over the baseline and are not novel and interesting enough to be publishable as-is.

Author Response

Dear reviewer,

       First, we would like to thank the reviewer for the positive and constructive comments and suggestions. We have substantially revised our manuscript after reading the comments provided by two reviewers. We will provide more details in the current version and please see the attachment.

Reviewer 2 Report

In this manuscript, the authors proposed a novel deep learning model of traffic speed prediction: fusing spatial-temporal graph convolutional network (FSTGCN). Generally, traffic flow is predicted by using traffic flow data, while traffic flow is predicted by using traffic speed data. However, there is a certain relationship between the traffic flow and the traffic speed in the road network. The authors integrate the practical relationship between the traffic flow and the traffic speed into the graph neural network, which is certainly helpful for the prediction accuracy. And the experimental results demonstrated that the FSTGCN can outperform other models for traffic speed prediction. This manuscript is interesting and innovative. Two comments are as follows.

(1)    The HA is a too old model, so it is recommended to delete it and try to supplement some updated traffic speed prediction models.

(2)    The matrix and vector should be italicized.

Author Response

      First, we would like to thank the reviewer and the editor for the positive and constructive comments and suggestions. We have substantially revised our manuscript after reading the comments provided by two reviewers. We will provide more details in the current version, and please see the attachment.

Reviewer 3 Report

In this paper, the authors fuse historical traffic flow with traffic speed data to predict future traffic speed with the proposed FSTGCN framework. However, I think the authors should carefully consider the following comments before the paper can be accepted for publication.

1、In section 4.4. Experiment Results and Analysis, it is better to give the definition of mean absolute error (MAE) and root mean squared error (RMSE) .

2、In section 4.4, the authors state poor performance of LSTM, however, from Table 2, we can clearly see that the RMSE performance of LSTM is the best among the evaluated algorithms and even better than the proposed method. Is there any error?

4、  The authors use reference [39] for the baseline algorithm ASTGCN, is it correct? 

5、Why the authors choose STGCN and ASTGCN as the baseline? are they the most typical and state-of-the-art? 

Author Response

Dear reviewer,

First, we would like to thank the reviewer for the positive and constructive comments and suggestions. We have some revised our manuscript after reading the comments provided yours. We will provide more details in the current version and please see the attachment.

Reviewer 4 Report

The proposal refers to a relevant practical problem concerning the vehicle traffic parameters prediction.

The relation between flows and speeds is obviously. The use of convolution and deep learning for traffic predication is well justified.

Introduction contains an excellent state-of-the-art.  The authors explain the reasons and the benefits of the research in the introduction chapter.

Preliminaries provides enough clear description of the involved formulas used int the proposed method.

Methodology explains the procedure linked to the related works.

What is less clear, why the new contributions lead to better results. Deep learning with weighted fuzzy logic rules can tackle the nonlinearity and the changing of the structure parameters too.

Experiments chapter are used to show that the proposed procedure gets better results compared to the previous methods.

Author Response

Dear reviewer,

      First, we would like to thank the reviewer for the positive and constructive comments and suggestions. We will provide more details in the current version and please see the attachment.